

# The diversity, evolution, and development of setal morphologies in bumble bees (Hymenoptera: Apidae: *Bombus* spp.)

Heather M. Hines[1,2], Shelby Kerrin Kilpatrick[2,3], István Mikó[2,4], Daniel Snellings[1,5], Margarita M. López-Uribe[2] and Li Tian[1,6]

[1] Department of Biology, Pennsylvania State University, University Park, Pennsylvania, United States
[2] Department of Entomology, Pennsylvania State University, University Park, Pennsylvania, United States
[3] Department of Entomology, Texas A & M University, College Station, Texas, United States
[4] Department of Biological Sciences, University of New Hampshire, Durham, New Hampshire, United States
[5] Division of Genetics & Genomics, Boston Children's Hospital, Boston, Massachusetts, United States
[6] Department of Entomology, China Agricultural University, Beijing, China

Corresponding author
Heather M. Hines, hmh19@psu.edu

## ABSTRACT

Bumble bees are characterized by their thick setal pile that imparts aposematic color patterns often used for species-level identification. Like all bees, the single-celled setae of bumble bees are branched, an innovation thought important for pollen collection. To date no studies have quantified the types of setal morphologies and their distribution on these bees, information that can facilitate understanding of their adaptive ecological function. This study defines several major setal morphotypes in the common eastern bumble bee *Bombus impatiens* Cresson, revealing these setal types differ by location across the body. The positions of these types of setae are similar across individuals, castes, and sexes within species. We analyzed the distribution of the two most common setal types (plumose and spinulate) across the body dorsum of half of the described bumble bee species. This revealed consistently high density of plumose (long-branched) setae across bumble bees on the head and mesosoma, but considerable variation in the amount of metasomal plumosity. Variation on the metasoma shows strong phylogenetic signal at subgeneric and smaller group levels, making it a useful trait for species delimitation research, and plumosity has increased from early *Bombus* ancestors. The distribution of these setal types suggests these setae may serve several functions, including pollen-collecting and thermoregulatory roles, and probable mechanosensory functions. This study further examines how and when setae of the pile develop, evidence for mechanosensory function, and the timing of pigmentation as a foundation for future genetic and developmental research in these bees.

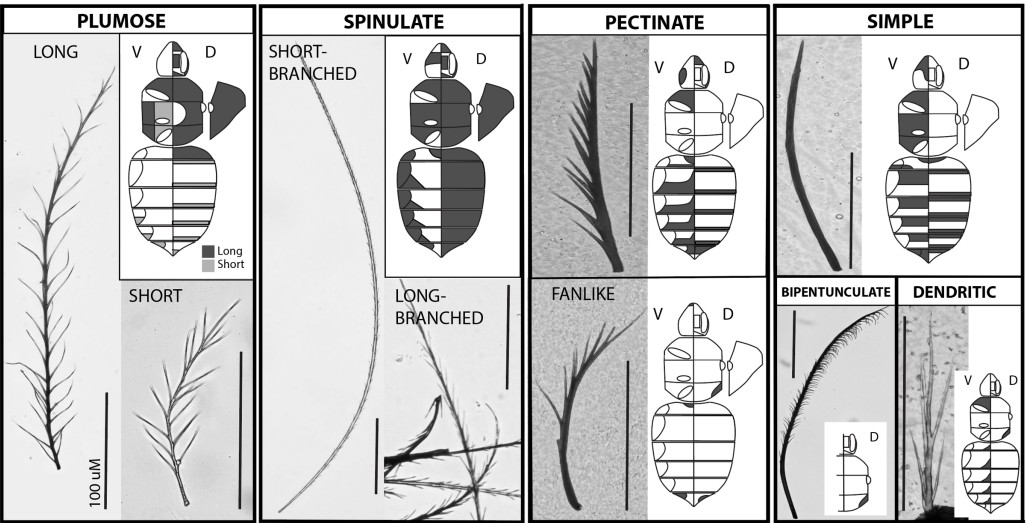

**Figure 1** **The major setal types on the bumble bee body.** Each major type occurs in a separate outlined box, with a close up photo of the type and a diagram of the body shading where that type occurs along the body. Plumose and spinulate have long and short subtypes shown. Images of the body include a diagram of the ventral side on the left half (leg positions and tergite lateral extensions outlined) and the dorsal side on the right (eye, face, and vertex regions outlined in the head), and on the far right shows a lateral view of the pleuron. Shading reflects where the setae of that type occur, although density levels vary. Bar = 100 uM.

# INTRODUCTION

From predaceous apoid wasp ancestors, bees (Hymenoptera: Apoidea: Anthophila) have evolved the innovation of feeding their young pollen. Pollen attaches to the setae (hairlike unicellular evaginations of the cuticle with a socketed base) of parts of the bee body while foraging on flowers and is subsequently groomed using leg combs into a pollen mass they feed their young (*Jander, 1976*; *Thorp, 1979*). The evolution of pollinivory coincides with the origin of branches on setae, a diagnostic synapomorphy for bees (*Michener, 2000*, Fig. 1). While these branches may have originally evolved to decrease air flow to prevent water loss or enhance pale coloration in ancestral xeric environments (*Michener, 2000*), they also increase the surface area of the setae and facilitate pollen capture (*Thorp, 1979*). The specific morphologies and densities of branched setae have been found to vary by taxon in a way that may facilitate specialization on certain pollen types (*Roberts & Vallespir, 1978*; *Thorp, 1979*; *Portman & Tepedino, 2017*).

Setae are clearly important to the bumble bees (*Bombus* spp.), as they exhibit a characteristically thick setal pile on their dorsal surface (dorsum) that makes them among the most pilous of the bees (*Roquer-Beni et al., 2020*). One potential role of these setae is to facilitate pollen collection. Bumble bees are key generalist bee pollinators in temperate zones of the northern hemisphere and must provide mass amounts of pollen to provision their young (*Goulson, 2010*). These bees demonstrate behaviors to maximize the accumulation of pollen on their bodies, which is then groomed *via* the legs into pollen baskets onto their hind metatibia (corbiculae) for transportation back to the nest (*Heinrich, 1979*; *Morse, 1982*; *Portman, Orr & Griswold, 2019*). Another potential role of

bumble bee pile is as insulation for thermoregulation. Unlike their closest tropical relatives, which are smaller and have considerably less pile, bumble bees are cold-adapted. The need in cold conditions to heat their bodies quickly for flight using mesosomal muscles (*Heinrich, 1979*), and to maintain heat, may benefit from insulative pile. *Church (1960)* found that depilated bumble bees lost heat through convection considerably more than those with pile, and that pile density and length impacted the degree of insulation. The importance of setae for thermoregulation is also supported by the relationships of setal length and density with geographic and climatic factors. Species from warm climates have shorter setae than species from cold climates (*Peat et al., 2005*), setal length correlates with elevation (*Peters et al., 2016*), and arctic bumble bees have denser setae on their heads (*Wilby et al., 2019*). Similarly, *Stiles (1979)* found that bumble bees are more sexually dimorphic in color and pile features in colder latitudes and altitudes, with males showing longer and less dense pile and lighter coloration than females, a likely adaptation to more extreme weather exposure given that males cannot return to thermostable nests.

Setae also serve the role of imparting the colors of these bees. Bumble bees exhibit exceptional diversity in color patterns on their dorsum, largely as a result of adaptive convergence and divergence onto numerous (>24) regional Müllerian mimicry complexes across their Holarctic and South American distribution (*Williams, 2007*). Setae are pigmented yellow, orange/red, black, brown, or white, with colors tending to vary by sclerites of the dorsum. For females, the resulting bright patterns are aposematic advertisements of their toxic venom, and males gain a mimetic advantage from matching them. Coloration of the pile may also serve thermoregulatory or other adaptive purposes as dark colors are more abundant in the tropics and in higher altitudes and latitudes and red colors are more common in higher altitudes (*Williams, 2007*). Black pile heats and cools faster than lighter colors (*Nixon & Hines, 2017*).

Bumble bee setal coloration has been a recent model for evolutionary genetics research. Colors of these setae have been found to result from shifts in melanin type (*Hines et al., 2017*) with the exception of yellow, which across bumble bees involves a novel type of pterin (*Hines, 2008b*) in addition to melanins (*Polidori, Jorge & Ornosa, 2017*). Developmental observations of these bees have revealed that melanization takes place at late pupal instars just prior to the shedding of pupal cuticle and yellow pigments enter setae after adult emergence (*Tian & Hines, 2018*). Using this knowledge of timing, developmental genomics across late pupal and early adult development in the bumble bee *Bombus melanopygus* has revealed how upstream Hox genes that impart segment identity (*abdominal-B*) ultimately trigger pigment divergences through downstream modification of various melanin genes (*Tian et al., 2019*; *Rahman et al., 2021*). Further research on color genetics in these bees could benefit from improved understanding of how and when setae of different coloration develop.

Several studies have led to a general synopsis on the formation of setae in holometabolous insects (*Keil & Steiner, 1991*; *Keil, 1997*; *Lawrence, 1966*; *Lees & Waddington, 1942*; *Walter et al., 1996*). Each insect seta—a cuticle-lined, non-porous hair-like protuberance of the exoskeleton—is developed by two epidermal cells, the trichogen (secretes setal cuticle) and tormogen (secretes socket membrane surrounding

setal base) cells (*Chapman, 1998*; *Keil, 1997*). Setae are sometimes innervated, in which case they have mechano- or chemosensory function. Innervation is detectable electrophysiologically by measuring electrical activity in response to setal deflection using single-cell recording (*Sutton et al., 2016*) or morphologically using TEM through the presence of the cuticle-bound, microtubule-rich dendritic ending of the sensory neuron, the tubular body (*Thurm, 1964*). Epidermal cells differentiate into setal cells shortly after apolysis of larval cuticle in pupae. The seta is initiated as an evagination of the trichogen cell membrane, grows to its terminal length and shape, and then begins to deposit cuticle on the outer surface. Folds in the epidermal surface promote the formation of longitudinal ridges on the setal cuticle and deposition of pigment inside the seta begins near the end of cuticle secretion. When cuticle deposition is complete, the cytoplasm of trichogen cells in the setal lumen reduces dramatically in size until it disappears completely (*Keil, 1997*).

While the bee setae are not generally thought to be innervated, it has been suggested that at least some setae of the pile may be potentially involved in mechanosensation. Bumble bees are receptive to electric fields, which aids in floral detection and decision-making while foraging (*Clarke et al., 2013*; *Sutton et al., 2016*) and setae of the pile could potentially play a role in this function. *Sutton et al. (2016)* showed that electrical pulses at the setal bases are generated in response to these electric fields. Although innervated setae with distinct tubular bodies have been associated with non-pile setae of the pronotal sensory plate of the honey bee and the flagella of males of some bumble bee species (*Thurm, 1964*; *Ågren & Hallberg, 1996*), morphological evidence for mechanoreceptivity, *i.e.*, the presence of tubular body, of setae of the bee pile has yet to be explored.

Variation in setal morphologies has been mentioned for bees more generally, especially in relation to pollen collecting, grooming using leg structures, and taxonomy (*e.g.*, *Thorp (1979*, *2000*)). A few authors have mentioned the general presence of different types of setae in bumble bee pile, mentioning some setae have longer branches and others have shorter ones (*Alford, 1975*; *Benton, 2006*). However, the diversity of types of setae in bumble bees and their distribution on the body has yet to be characterized. Given the ecological and evolutionary importance of these setae to these bees and that setal characters (color, length) are often used for species-level identification in this group, more specific descriptions of setal morphology are needed. A framework for identifying the types of setae that are present and where they are distributed on bumble bees will enable understanding of the relative biological roles of setae in these bees.

To facilitate evolutionary, genetic, and functional research, here we analyze the morphology, development, and evolution of bumble bee setal color and branching types. First we examine and characterize the different setal types present across the bumble bee *B. impatiens*, determining the location of each setal type on the body and how these differ by caste and sex. We then take the two most common setal types and characterize their distribution on the body across the phylogeny, sampling 127 of ~260 (*Williams, 1998*, updated online) species of bumble bees. These data are used to assess whether certain ecological factors may relate to and thus explain the adaptive value of these setae, to understand the origin of these setal types, and to assess the degree to which setal morphology may be useful as a diagnostic character at different phylogenetic scales. As a

foundation for future genetic research on these bees, we provide data on how bumble bee setae develop, tracking the timing and morphology of the first growths of these setae using microscopic techniques, and more precisely examine the timing of pigmentation of both melanin and pterin pigments in newly eclosed adult bees.

## MATERIALS AND METHODS

### Setal morphology

To examine setal types across the body, 26 adult *Bombus impatiens* Cresson, including 16 workers and 10 queens, were collected from two lab-reared colonies (Koppert Biological Systems, Howell, MI, USA). Additionally, one wild-caught male and four wild-caught worker specimens of *B. impatiens*, collected in Centre County, Pennsylvania, USA were included in the analysis. Individuals were euthanized, pinned, and stored in a −20 °C freezer to maintain pliability prior to examination.

Setae were shaved from the surface of each body region (tergites, sternites, subregions of the mesosoma and head, and legs) individually using a disposable scalpel. Setae from each segment were slide-mounted on glass slides with plastic coverslips and sealed with clear nail polish. Slide-mounted setae were observed using Olympus UPlanFLN 4X UIS2 and LMPlanFLN (10×/0.25; 20×/0.40 and 50×/0.50) UIS2 objectives, fitted on an Olympus CX41 microscope. Setae across the body were categorized into phenotypes considering terminology and characters for describing setal morphology from multiple arthropod and plant resources (*e.g.*, *Garm & Watling (2013)*, *Larsen (2003)* and *Harbach & Knight (1980)*). Hymenoptera specific terms used in this work were matched and aligned to the Hymenoptera Anatomy Ontology (HAO, *Yoder et al., 2010*).

After characterizing setae from shaved material, additional bee specimens were examined *in situ*, to determine specific locations of setae and consistency across specimens. Specimens were observed using an Olympus SZX16 dissection microscope with an Olympus SDF PLAPO 1XPF objective, under 7–115× magnification or on an Olympus SZ61 microscope up to a 45× magnification. To observe setal morphologies properly, magnification typically has to be >30×, specimens should be dried as opposed to observed in ethanol to enable better focus (pinned or dried from ethanol), and sometimes moving setae with a pin was required to examine layers. Setal type images were taken manually with a Canon EOS 70D camera, mounted on the Olympus CX41 microscope, and aligned and stacked with Zerene Stacker (Version 1.04, Build T201711041830). Scanning Electron Microscope images of the two main setal types *in situ* were taken from the face just below the antennae in *Bombus bimaculatus* Cresson (Urbana, IL, USA) using 574× SEM magnification. Although this is a different species from *B. impatiens*, these setal types are similar across bumble bees, thus this represents the general differences across bumble bees.

### Variation in setal type distribution along the phylogeny

The two most prevalent types of setae found in bumble bees are spinulate and plumose (Fig. 1). We assessed the distribution for these two setal types along the dorsum in females of approximately half of the bumble bees species (127 of ~260 species; Datafile S1) covering all subgenera in a fairly even distribution across the phylogeny (*Hines, 2008a*). This

sampling was chosen to allow a sense of the extent of phylogenetic signal in this phenotype and its promise for taxon diagnosis. For most species, 2–3 individuals were sampled, usually from different populations, which allows a better sense of intraspecific variation. However, we only included one specimen for 50 of the species because of limited numbers of samples available. Most individuals sampled were workers, although occasionally queens were also included. For *Psithyrus* social parasites, only females were examined. Queen phenotypes matched closely those of workers. We examined the setal types of the vertex and the face, dorsal and lateral regions of the mesosoma, and the dorsal and lateral regions of the visible metasomal tergites (T1–T5). Tergite 6, which contains the stinger, was not included due to there being a predominance of other setal types besides spinulate and plumose in this region and this region generally showing considerable baldness. Both long-branched and short-branched spinulate setae were lumped together as were short plumose and long plumose setae given the continuous nature of these distinctions.

For each individual the distribution of the plumose setal type was diagrammed (spinulate setae occur across all of these segments in nearly all species) on a bee template. From this, we categorized the amount of plumose setae per body region into the three groups as follows. MARGIN: plumose setae occur only along the posterior margin of the metasomal tergites usually as a single or double row of setae. <25%: plumose setae occupy more than just the marginal region but occur in less than 25% of the area of the tergites. 25–50% and >50%: plumose setae occur in 25–50% or >50% of the area of the tergites, respectively. Further distinctions above 50% are not made as plumose setae tend to expand towards the anterior of the tergites and the anterior portion is often not fully visible to discern the full extent of plumose setal distribution. These designations refer only to the area in which the setal type can be found and not to the density of these setal types, which can be quite variable by taxon. Notes were taken in cases where setal density appeared unusually high or low.

Using the phylogeny and these data, we examined the degree of phylogenetic signal in average plumosity by species and by more specific body region to examine which parts of the body show the greatest phylogenetic conservation. To test for phylogenetic signal, we estimated Blomberg's K using the function *phylosig* in the R package 'geiger' (*Harmon et al., 2008*). The Blomberg K parameter quantifies the amount of trait variation within and among clades to test whether the amount of phylogenetic signal in the data exceeds the quantity of signal expected by random chance (*Blomberg, Garland & Ives, 2003*). K-values close to 1 indicate strong phylogenetic signal.

We assessed statistical significance through randomizations by comparing the number of times our simulated values of K were different than our observed values. For traits where we found phylogenetic signal, we fitted five phylogenetic models (Brownian motion, Pagel's lambda, Ornstein–Uhlenbeck, white noise and early burst) and used the Akaike information criterion (AIC) to examine which models of evolution better fit the data. All analyses were done in R version 4.2.0 (*R Core Team, 2022*).

## Setal development

Setal development was examined using an Olympus Confocal Laser Scanning Microscope (CLSM) and standard compound microscopes. For this, developing pupal tissues of the growing adult epidermis from stages P2–P7 (*Tian & Hines, 2018*) were excised from metasomal tergites (spinulate setae) and the mesosomal dorsum (mostly plumose setae) using dissections in cold phosphate buffer solution (PBS) and removal of non-epidermal tissues as much as feasible for the tissue type and stage. These were examined for patterns of setal branching and growth across development. An Olympus BX51 compound microscope with a long working distance 40X objective lens with the sample placed in PBS or glycerol was used for imaging.

## Setal innervation

We examined cellular patterns at the setal base and whether the tubular body was present using CLSM and Transmission Electron Microscopy (TEM) on metasomal tergite 2 of freshly emerged adults of *Bombus impatiens* (callows). For CLSM, specimens were dissected in PBS and placed on a PBS droplet in between two #1.5 coverslips and examined with an Olympus FV10i CLSM at the Microscopy and Cytometry Facility at the Huck Institute of Life Sciences, Pennsylvania State University. We used two excitation wavelengths (473 and 559 nm) and detected the autofluorescence using two channels with emission ranges of 490–590 nm and 570–670 nm. The resulting .oib files were assigned pseudo-colors (red and green) corresponding to the emission wavelengths. Volume-rendered micrographs were created using FIJI (Schindelin et al. 2012). For TEM, specimens were dissected in 0.1M cacodylate buffer and fixed in 2.5% glutaraldehyde in 0.1M cacodylate buffer with 5% sucrose for 2 h at room temperature and 24 h at 4 °C and washed in 0.5M cacodylate buffer. Fixed specimens were stained with osmium tetroxide for 2 h and uranyl acetate for 12 h, dehydrated through an ethanol series, and embedded in eponate. Blocks were trimmed and sectioned using a Leica UCT ultramicrotome. Sections were collected on grids and then double-stained with lead citrate and uranyl acetate. Sections were imaged with a JEOL 1200 TEM at the Microscopy and Cytometry Facility at Penn State. TEM micrographs were further processed using Adobe Photoshop CS5.1.

## Setal pigmentation

Our previous observations using visually apparent differences revealed that in black and red setae, deposition of setal pigment begins around Stage P13 (Partially melanized with appendages blackening, ~6 days post larval; pupal staging information available in *Tian & Hines (2018)*) pupae, whereas pigmentation of the yellow setae does not begin until a few hours after adult emergence from the pupal cocoon, and that setae of all colors do not complete coloration until 1–2 days after eclosion (*Tian & Hines, 2018*). Here we more precisely examine the temporal deposition of pigmentation in the hours after adult eclosion. For black setae we examined development of melanic pigmentation at the single seta level in workers of *B. impatiens* at 0, 4, 8, 12, 16, 20, 24, and 72 h post adult emergence from cocoons. For this we imaged individual bee setae using a standard light set-up under an Olympus BX51 microscope at 200× magnification and measured percentage darkness

of developing black setae relative to 3-day old adults. For yellow setae, we shaved setae from the lateral regions of the mesosoma from *B. impatiens* workers, sampling 3–4 replicates each of the following numbers of hours post-adult emergence from cocoons: 0, 12, 16, 20, 24, 28, 32, 48, and 72 h. Setae were placed in 1.5 ml centrifuge tubes along with 20 uL of 0.01M sodium carbonate ($Na_2CO_3$) and vortexed for 10 s, a protocol that effectively removes the yellow pterin pigment without modifying its chemical properties (*Hines, 2008b*). Each sample was spun down in a minifuge for 15 s, heated in the thermomixer at 81 °C at 900 rpm for 1 min, and chilled on ice for 1 min, and this heating and cooling process was repeated two more times. Samples were then run on the Nanodrop spectrophotometer using 2 ul of solution and scanning across UV and visible wavelengths (200–900 nm). Absorbance quantities were measured from the spectrum as the difference in absorbance between the absorption peak at 390 nm and the trough between 320 and 390 nm. While most bees were of similar size, to correct for differences between the amount of setae present on the episternum by body size, marginal wing cell measurements were taken under 0.67× magnification on an Olympus SZ61 stereo microscope and absorbance values were corrected using the following equation: (measured absorbance)/(marginal cell length/average overall marginal cell length) = corrected absorbance value.

Images were obtained of black and yellow setae from 0 h newly emerged adults to fully developed adult coloration using a Nikon E600 light microscope with Nomarski and fluorescent optics at 10–100× power and photographed using a SPOT Insight digital camera attachment and associated software.

## RESULTS

### Setal morphology

We recognize the following types of setae in *B. impatiens*, which are diagrammed in Fig. 1 and tabulated in their key features in Table 1:

1. **Spinulate—Short-branched:** Long rachis (av. = 0.8 mm in worker) of consistent width with short, straight, branches that are spine-like (Fig. 2). Branches run along the entire rachis length and are usually paired but staggered in position so as to have circumrachis distribution. Branches decrease in size near the distal end of the rachis, and angle towards the tip of the rachis at approximately 20°. Tip of rachis abruptly pointed. This term follows *Harbach & Knight (1980)* mosquito vesture nomenclature.
   **Long-branched:** Same as with short-branched but with longer branches along the entire length and the tip of the rachis less abruptly pointed. Shows intermediacy between short-branched spinulate and long plumose setae. This term follows *Harbach & Knight (1980)* mosquito vesture nomenclature.

2. **Circumplumose "Plumose"—Long:** Long rachis (av. = 0.6 mm in workers) of continuous width with long, curved/wavy, randomly arranged branches along the proximal two thirds to three fourths of the seta. Remaining length of seta possesses more regularly paired sets of bilateral branches. Branches decrease in size towards the distal, pointed tip, and are angled towards the tip of the rachis at approximately 30°. Overall
**Table 1 Features of each designated setal type in bumble bees.**

| Setal type | Branch location | Relative branch length | Branch angle from rachis | Rachis length | Rachis shape | Other branch characters |
|---|---|---|---|---|---|---|
| Spinulate—short branched | Circumrachis | Short | 20° | Long | Straight | Straight, spinelike |
| Spinulate—long branched | Circumrachis | Medium | 20° | Long | Straight | Straight, spine to hairlike |
| Plumose—long | Circumrachis | Long | 30° | Long | Straight | Tip curvature |
| Plumose—short | Circumrachis | Long | 40° | Short | Straight | Tip curvature |
| Simple | No branches | | | Short-medium | Straight or curved | |
| Pectinate—standard | One-sided row | Variable | | Variable | Straight | Thicker |
| Pectinate—fine | One-sided row | Usually long | | Variable | Straight | Finer |
| Pectinate—fan-like | One-sided row | Variable | | Short-Medium | Curved | Usually thicker |
| Bipentunculate | Two bilateral rows basally rotate to one side apically | Short | Nearly 90° | Medium | Curved | Thick, apical curvature, dense |
| Dendritic | Circumrachis | Long | 20° | Short | Straight | |

**Note:**
Branch length is relative to the rachis.

setal length tends to be shorter than spinulate setae. **Short:** Same as for long plumose but shorter (av. = 0.15 mm in workers) and with angles of branches closer to 40°. There can be a continuum in lengths from short to long plumose. This follows mosquito (*Harbach & Knight, 1980*) and crustacean (*Larsen, 2003*; *Garm & Watling, 2013*) terminology and is more broadly used to refer to feather-shaped hairs.

3. **Simple**—Slender rachis, with seta wider basally and gradually narrowing into a pointed tip. Lacking branches. Appearing straight when very short, but may have a gentle curved shape when longer. Length varies but tends to be around 0.2 mm in workers. This term is applied broadly for this type of seta.

4. **Pectinate—Standard:** Robust, short to long depending on location, near-continuously thick rachis, and straight, with robust branches primarily distributed along one side of the seta. Often branches decrease in length towards apex. One to a few branches may be present randomly along the opposite side. **Fine:** Same as standard pectinate but slender with seta wider basally and gradually narrowing into a pointed tip, with thin, long branches primarily distributed along one side of the seta. **Fan-shaped:** Same as standard pectinate but curved, thus splaying in a fan-like fashion. Branches may be short (appearing as a serrated texture) or long. This term is consistently used in this same way across the literature (*e.g.*, *Harbach & Knight (1980)*; *Larsen (2003)* and *Garm & Watling (2013)*).

5. **Bipentunculate**—Robust, long (~0.4 mm in workers), near-continuously thick rachis with two rows of closely-set branches along one side of the terminal three quarters of the rachis, that bend towards each other along with rachis curvature near the tip to create a more close-set folded double row, giving it a comb-like or eyelashes-like appearance.

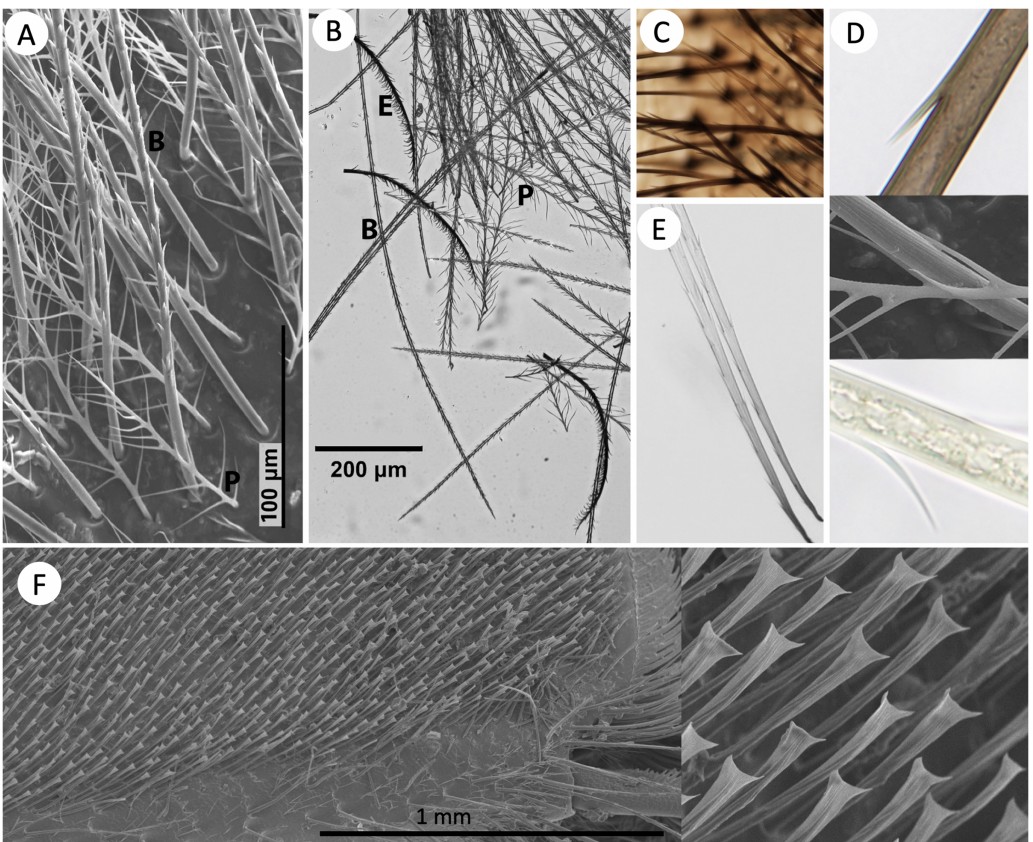

**Figure 2 Images of setal morphology diversity and pigmentation.** (A) SEM image of *B. bimaculatus* setae located on the face, just below the antennae, showing both spinulate (B) and plumose (P) hair types. (B) Light microscope image of shaved setae from the posterior-lateral thorax of *B. impatiens*, showing spinulate, plumose (long and short), and bipentunculate (E) setal types. (C) Simple setae (lighter colored) on the metasomal sternites of *B. impatiens*. (D) Magnification of the black (top) and yellow (bottom) setae of *B. impatiens* when near full pigmentation. Middle: SEM image showing fine ribbing on setae (*B. bimaculatus*). (E) Black setae of *B. impatiens* imaged at the 0hr callow stage showing how color progresses from the base of the seta towards the tip and how percent darkness can be assessed in these hairs using image data. (F) Keirotrichia of the posterior (inner) side of the hind tibia with a close-up view of these setae on the right.               

The apex of the branches curve downwards towards the rachis base. First quarter of the rachis with no outgrowths. Tip of rachis pointed. "Pentunculate" is applied to mosquitos (*Harbach & Knight, 1980*) to refer to a one-row comb-like structure, thus we use bipuntunculate to apply this to a double row.

6. **Dendritic**—Especially short (~0.1 mm in workers), with continuously wide rachis, terminating in a long, pointed tip. May be straight or curved. Long, straight branches radiate from the rachis with lengths that are a large fraction of the total length of the rachis and thus give the appearance of arising from the base of the seta. Branches may be randomly arranged or in bilateral pairs. Branches angle away from the rachis at approximately 20°. "Dendritic" is applied mosquitos (*Harbach & Knight, 1980*) to any tree-like structure involving a basal stem with longer branches radiating from it.

Figure 2 shows the morphology of the main setal types when mixed, in both SEM and shaved microscope samples. At a very close microscopic level, grooves are evident on the setae (Fig. 2D). These are much larger in spinulate setae than in plumose setae.

## Distribution of setal types in *B. impatiens*

The distribution of setal types in worker *B. impatiens* varies between body regions and segments (Fig. 1, raw data in Datafile S2). The two most prevalent types of setae on the dorsum of the body in *B. impatiens*, and bumble bees in general, are short-branched spinulate and long plumose. Being somewhat shorter, long plumose setae tend to make a downy underlayer, thus creating an especially dense appearance of pile on the dorsal and lateral surface of the mesosoma, along the vertex of the head, on the face around the antennal base, and in the densely setose lateral corners of the first metasomal tergite. The second to fifth dorsal metasomal tergites are covered almost exclusively by spinulate setae. These two setal types also occur in the ventral regions of the mesosoma, although plumose setae are shorter in the region between the wing bases. Spinulate setae also occur laterally on metasomal sternites.

The other setal types tend to have more restricted distribution. Simple unbranched setae can be found along the anterior regions of tergites but are generally not visible unless the tergites are fully protracted (Fig. 1). They also are more common setae on ventral body regions: anterior portions of metasomal sternites (Fig. 2C) and throughout the ventral regions of the mesosoma. Pectinate setae, particularly fine pectinate, tend to dominate the remaining posterior portions of the metasomal sternites and parts of the head and anteroventral mesosomal regions. Several especially short setal types can be found along the posterior edges of the metasomal tergites including a mixture of simple, fine pectinate, short plumose, and dendritic, mixed with longer and easier to observe pectinate and long-branched spinulate setae. The sixth metasomal tergite, tends to have shorter and variable seta types, exhibiting a mix of simple, pectinate, and short plumose seta types along with a comparatively low density of longer spinulate setae. Bipentunculate setae are only present posterodorsally on the metanotum of the mesosoma, just behind the wings' point of attachment (Figs. 1 and 2B).

We focused primarily on the setal types of the bumble bee head, mesosoma and metasoma, however we also performed a cursory examination of types of setae elsewhere in the body. Setae of legs vary considerably by leg segment and region, with all types of setae except for bipentunculate in various locations. In addition, legs exhibit some different setal types, including keirotrichia which are thought to be used to groom the wings (*Michener, 2000*; *Rozen, 2021*), which in these bees are simple setae with a blunt squared off end with slight bifid expansion in each corner of the square to make a slightly triangulate appearance (Fig. 2F). These make a uniformly spaced, even iridescent velvety coat on the distal hind posterior (inner) femur and across the surface of the posterior (inner) hind (back side of the corbicula) tibia. Also, among the setae framing (to the left and right of) the corbicula are serrated pectinate setae where branches are one-sided, very short, and tend to be located near the tip of the rachis. In general the corbicula lacks setal branches on the side facing the corbicular load, with setae around the corbicula being pectinate

branched or simple setae with branches, when present, facing away from the center. The legs also bear patches of short plumose setae at the leg joints, with especially pronounced ones near the metatibial-metatarsal junction. They also have several larger setae/bristles along margins between the segments of the legs and along the surface of some leg segments. The antennae are covered with a fine velvet of very short simple setae (cf. *Ågren & Hallberg, 1996*), with the scape the only antennal segment bearing longer setae which include simple and pectinate, similar to metasomal sternites. The setae on the wings and female and male genitalia are simple.

The types of setae and their distribution are conserved among examined individuals of the same caste. Setal types also are very similar in their distribution between worker and queen castes. Observed differences between worker and queen *B. impatiens* are that queens have (1) distinctly fewer short setae along the posterior margins of the tergites, (2) spinulate setae with longer branches on what we refer to as the "ventral" side (also referred to as the posterior by *Michener (2000)*, includes the occiput, hypostoma, and genal area) of the head, (3) denser setae on the ventral side of the head, and (4) there are more fine pectinate setae medially along the posterior margins of the sternites. The distribution of the most common setal types are also very similar between workers and males. Observed differences between worker and male *B. impatiens* are that males have (1) increased setal density on the face, (2) simple and short plumose setae present on the base of the mandible (only simple setae are present in workers and queens), (3) long plumose setae mixed in the area of long-branched spinulate setae on the ventral side of the head, (4) conspicuously more setae and longer setae on the sternites, particularly long-branched spinulate and fine pectinate, (5) some long plumose setae present on the posterior margins of the sternites, (6) fine pectinate setae present more densely and completely across the posterior margins of the sternites, and (7) Metasomal sternite 7, which is only present in males, possesses pectinate, long-branched spinulate, fine pectinate, and short plumose setae.

## Distribution and phylogenetic signal of the plumose and spinulate setal types in bumblebee taxa

For all bumble bees examined across the phylogeny, the primary setal types in the dorsum are long plumose and spinulate setae, thus we focused our cross-species comparisons on these two setal types. Nearly all species displayed spinulate setae throughout the dorsum, although for a few of the more plumose species, especially in the posteriormost tergites (4th and 5th) nearly all setae were plumose and spinulate setae were absent. While in most cases setae could be assigned to discrete types, for some species, branches were longer than others, making distinction between spinulate and plumose less clear. In some *Bombus s.s.* (*hypocrita* Perez, *patagiatus* Nylander, *moderatus* Cresson), Old World *Pyrobombus* (*biroi* Vogt, *flavescens* Smith, *cingulatus* Wahlberg), *Thoracobombus* (*exil* (Skorikov), *veteranus* (Fabricius), *weisi* Friese, *humilis* Illiger), *Subterraneobombus* (*melanurus* Lepeletier, *fragrans* (Pallas), *subterraneus* (Linnaeus)) and *Sibiricobombus* (*sulfureus* (Fabricius)) the branch length decreases from posterior to anterior in some tergites, especially marked in metasomal tergite 2, making the line between plumose and spinulate more nebulous. When plumose setae occur on the posterior margins or tergites ("margin" in our coding),

often these just occur as a few short light setae in single or double row distribution, thus can be very low in number for this designation.

Across all bumble bees, the mesosoma and head consistently showed a high density of plumose setae (Fig. 3) along with spinulate setae, making for a thick downy underlayer. This is especially dense on the lateral regions (lateral parts of pleuron covered with pile) and the mesonotum near the wing bases and along the dorsal mesosomal peripheral rim, and is less dense medially on the mesoscutum where for some species they were missing and hairs overall appear less dense. All bumble bees exhibit >50% of the area of these regions with plumose setae and most exhibit >90%. The only exception is the sister lineage to the rest of the bumble bees, *Mendacibombus*, which exhibited the least plumose setae on the mesosoma and head, with one species, *B. handlirschianus* Vogt, even demonstrating <50% plumose setae due to absence on the median region of the mesoscutum and lateroventral mesosomal regions and much of the face.

The metasoma in bumble bees generally exhibits less plumose setae, showing considerable variation in location of these setae (Fig. 3). Evolutionary increases of plumose setae tends to involve increase from posterior to anterior on metasomal tergites (Fig. 3, inset). When plumose setae do occur in the metasoma, it tends to be less dense. Some notable exceptions are that T1 (the first metasomal tergite) often has dense plumosity, especially on lateral portions, and some species have nearly exclusively or majority plumose setae in posterior tergites, *e.g.*, *Pyrobombus: remotus* (Tkalcü), *infrequens* (Tkalcü), *pratorum* (Linnaeus), *Thoracobombus: zonatus* Smith, *pseudobaicalensis* Vogt, *tricornis* Radoszkowski, and *trinominatus* Dalla Torre. *B.* (*Megabombus*) *longipes* Friese has a lower density of plumosity but lack the spinulate setae in much of the tergites, thus generating "darker" parts of the tergites that have lower setal density and show the black cuticle through, similar to honey bees (*Apis mellifera* L.).

When examining averages by metasomal region (Fig. 3, Datafile S2), the first and fifth metasomal tergites have the most plumose setae, followed closely by the fourth tergite, and with the least plumose setae in the second followed by the third tergite. There is low cross species variance in the mesosoma (0.01) but fairly similar variance by metasomal tergites (1.88 [T2] – 2.42 [T5]) across species. These patterns are congruent with estimates of phylogenetic signal based on Blomberg's K (Table 2). We found a lack of phylogenetic signal for the traits in the head and mesosoma, which is expected because they nearly all were >50% plumose. Setae patterns in T1–T5 showed strong phylogenetic signal ($p < 0.001$) indicating that more closely related species show more similar setal type patterns than species that are more distant phylogenetically. Phylogenetic signal was highest in T1, T4, and T5, as indicated by effect size K, and was especially strong when averaging across the body.

Overall plumose setal distribution shows fairly strong and significant phylogenetic signal at the species group or subgeneric level (Fig. 3). There is a general absence of these plumose setae and the least amount of plumosity across the body overall, in the sister lineage to the rest of bumble bees, *Mendacibombus*, and low amounts in the next diverging clade, *Bombias*. The greatest amounts of plumose setae can be found in some of the longest tongued subgenera of the long-tongued *Bombus* clade (*Thoracobombus—Orientalibombus*),

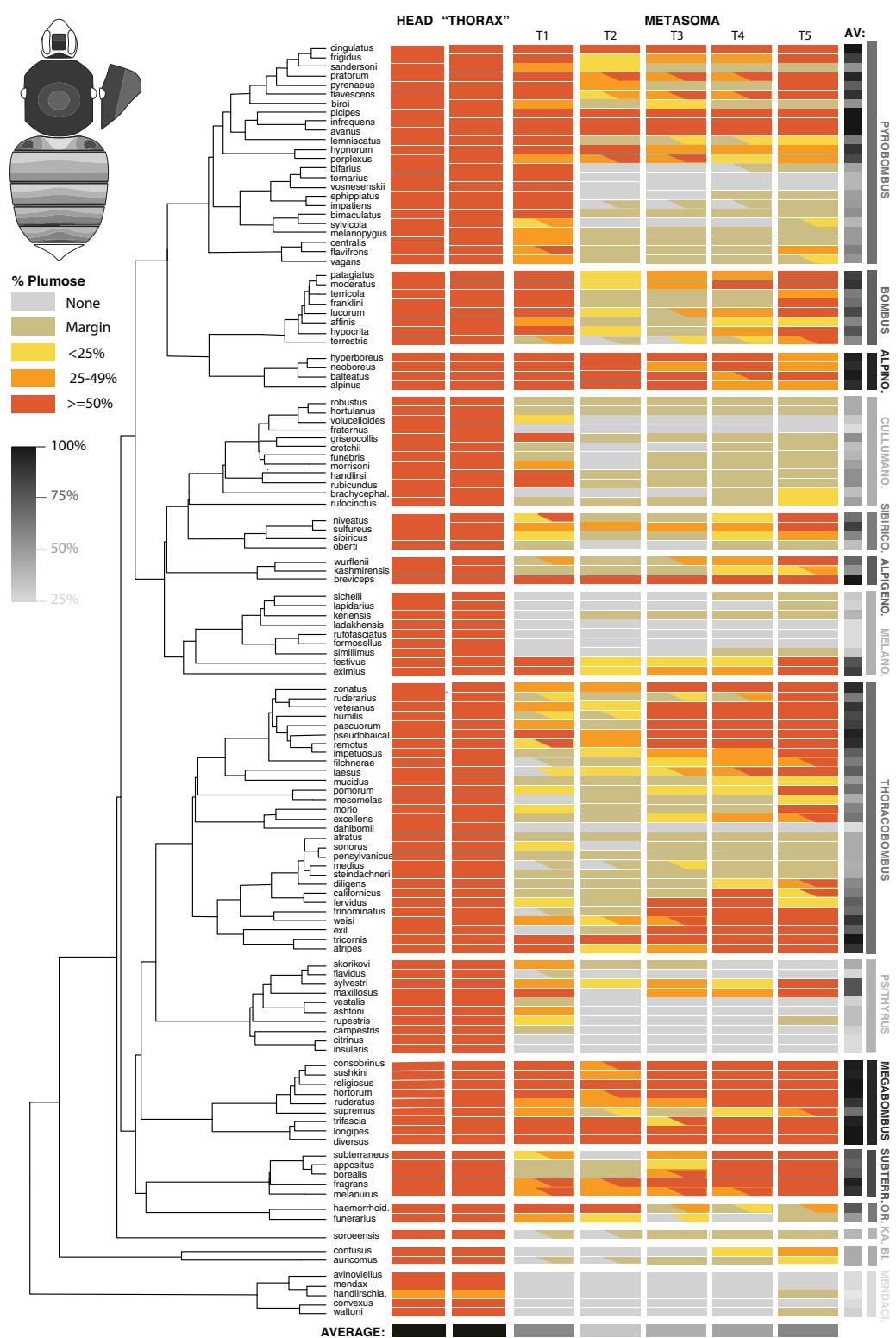

**Figure 3 The distribution of plumose setae on the head, mesosomal, and metasomal tergites in females in representative species (*n* = 127/~260) across the phylogeny of bumble bees.** This shows the percentage of plumose setae relative to spinulate setae in each region. Only metasomal tergites 1–5 are shown. Most codes are based on 2–3 individuals per species and are based largely on worker phenotypes.
**Figure 3** (continued)
Intraspecific variation is represented by partial codings for respective sections. The overall percent of the body with >50% of plumose setae for each species and from each segment combined across species is represented in grayscale. The bee at the upper left shows the directionality of increased plumosity for each segment, showing how when plumose setae are of low percent they occur in the posterior of the tergite and when percentages increase, plumosity extends towards the anterior, especially in the middle.

**Table 2 Phylogenetic signal based on Blomberg's K.**

| Trait | K | *p*-value | Best model |
|---|---|---|---|
| Head | 0.154 | 0.704 | – |
| Thorax | 0.154 | 0.699 | – |
| T1 | 0.888 | <0.001 | Pagel's λ |
| T2 | 0.672 | <0.001 | Pagel's λ |
| T3 | 0.625 | <0.001 | Pagel's λ |
| T4 | 0.745 | <0.001 | Pagel's λ, Brownian Motion |
| T5 | 0.641 | <0.001 | Brownian Motion, Pagel's λ |
| Average | 0.904 | <0.001 | Brownian Motion, Pagel's λ |

Note:
Statistical significance (*p*-value) was quantified based on 10,000 simulations. Best fitted models of trait evolution were determined based on the Akaike Information Criterion (AIC).

with highest amounts in *Subterraneobombus* and *Megabombus*. Higher amounts are also found in some of the subclades of the long-tongued *Thoracobombus*, most notably the Old World *Thoracobombus*, but this shifts to low plumosity in some of the warmer adapter New World *Thoracobombus* (*Fervidobombus* subclade). The short-tongued clade, which comprises the upper half of the tree in Fig. 3 (*Melanobombus—Pyrobombus*), generally shows lower amounts of plumose setae with the exception of a few clades. The exceptionally cold-adapted *Alpinobombus* has high plumosity across the metasoma as do many of the Old World *Pyrobombus*. In contrast, *Melanobombus* shows very low metasomal plumosity, as does New World *Pyrobombus*, and the *Cullumanobombus*. The socially parasitic *Psithyrus* have thick plumosity in the mesosoma but most of them lack much plumose setae in the metasoma.

Additional phylogenetic trends are apparent in the distribution of plumose setae across these regions (Fig. 3). The [*Pyrobombus, Bombus s.s., + Alpinobombus*] clade tends to have a lot of first tergite plumosity whereas other lineages are less prone to have plumose setae in this region, with *Thoracobombus* being more prone towards expansion in the posterior regions. *Orientalibombus* is prone towards more plumose setae in T2 than other lineages so has more anterior plumosity. Species also vary in the extent of plumose setae on margins. In New World *Thoracobombus* and *Cullumanobombus* the margins tend to have an edge of sparse, shorter plumose setae more than other lineages.

Intraspecific variation in these phenotypes is generally low within species, with most variation involving slight increases or decreases in the extent of plumosity in given segments that can push thresholds into adjacent plumosity group bins. However, in a few

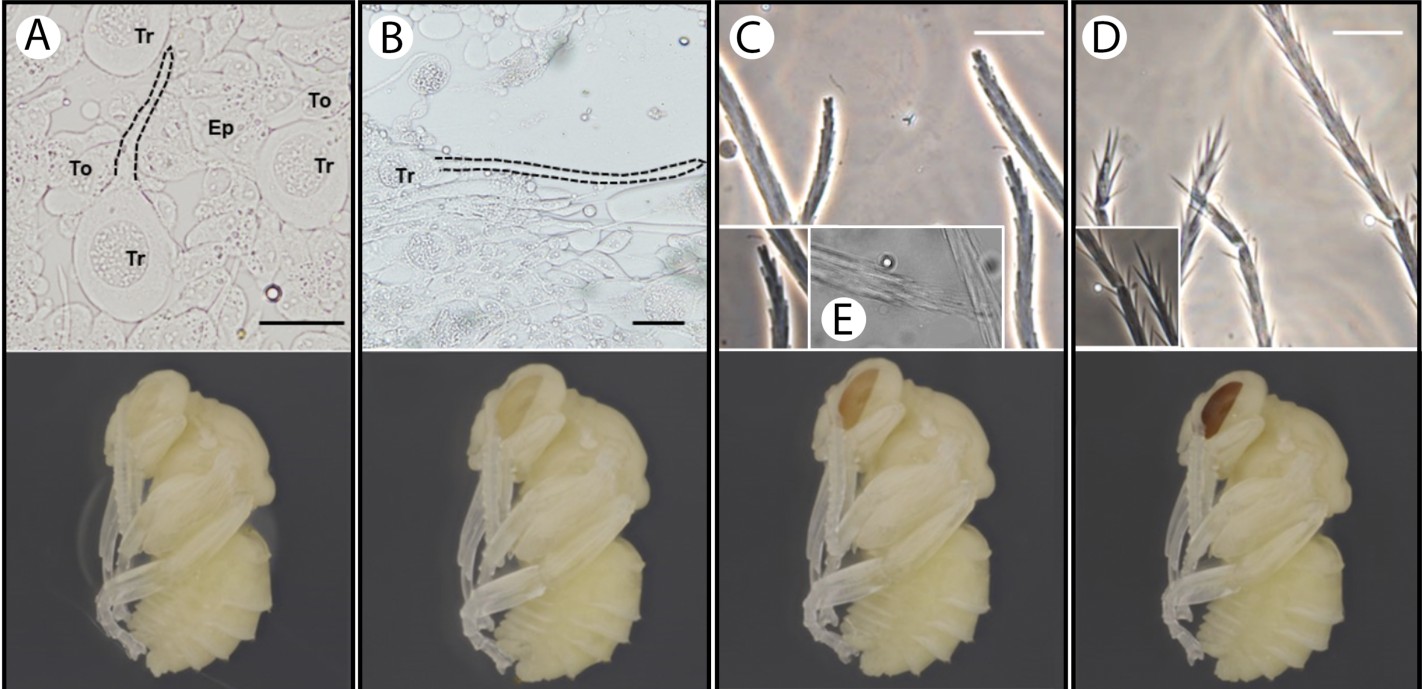

**Figure 4 Temporal patterns of developing setae in the bumble bee *B. impatiens*.** (A–D) Stage P2, P4, P5 and P7, respectively, with setal morphology for each stage diagrammed at top and a pupal image for the respective stage at the bottom. Tr, trichogen; To, tormogen; Ep, regular epidermal cells. Dotted lines outline the hair shaft. Small figures in (C) and (D) represent zoomed in views of spinulate setae, showing blunted branches at P5 and pointed branches at P7. The small figure in (E) represents the appearance of the tips of plumose mesasomal setae around P5. All samples except (E) are prepared from metasomal tergite 2, where spinulate setae are present. Scale bar = 40 μm.

species we noted distinct differences across populations, such as those in *B. terrestris*, *B. trifasciatus*, and in *B. haemorrhoidalis*.

## Setal development

Our observations revealed the presence of membrane evaginations on the trichoid cells (apical sprout of the trichogens) at stage P2 pupa (white-eyed, ~8 h after pupal initiation (*Tian & Hines, 2018*)) (Figs. 4A and 4a). After this stage, the seta continues to elongate (Figs. 4B and 4b). Setal branches appear at pupal stage P4 (peach eyed, 1.5–2 days after prepupal initiation). Early stages of setal branches have blunt ends for both plumose and spinulate setae (Figs. 4C, 4c and 4e). During this time (P4–P6), there is a higher density of branches near the tip of the seta suggesting that distal rachis elongation continues after branch initiation. The plumose setae have longer branches than spinulate early in branch formation, and in mid-development setae exhibit a bushy cluster of branches near the tip that will eventually separate more with the distal growth of the rachis (Figs. 4c and 4e). Cuticle deposition on the seta appears to take place around P6–P7 (Red-to-maroon eyed stage, ~2–4 days into pupation), where the ends of setal branches become pointed (Figs. 4D and 4d). It appears that such morphological transformation (blunted to pointed end) signifies the end of elongation of both the rachis and setal branches.

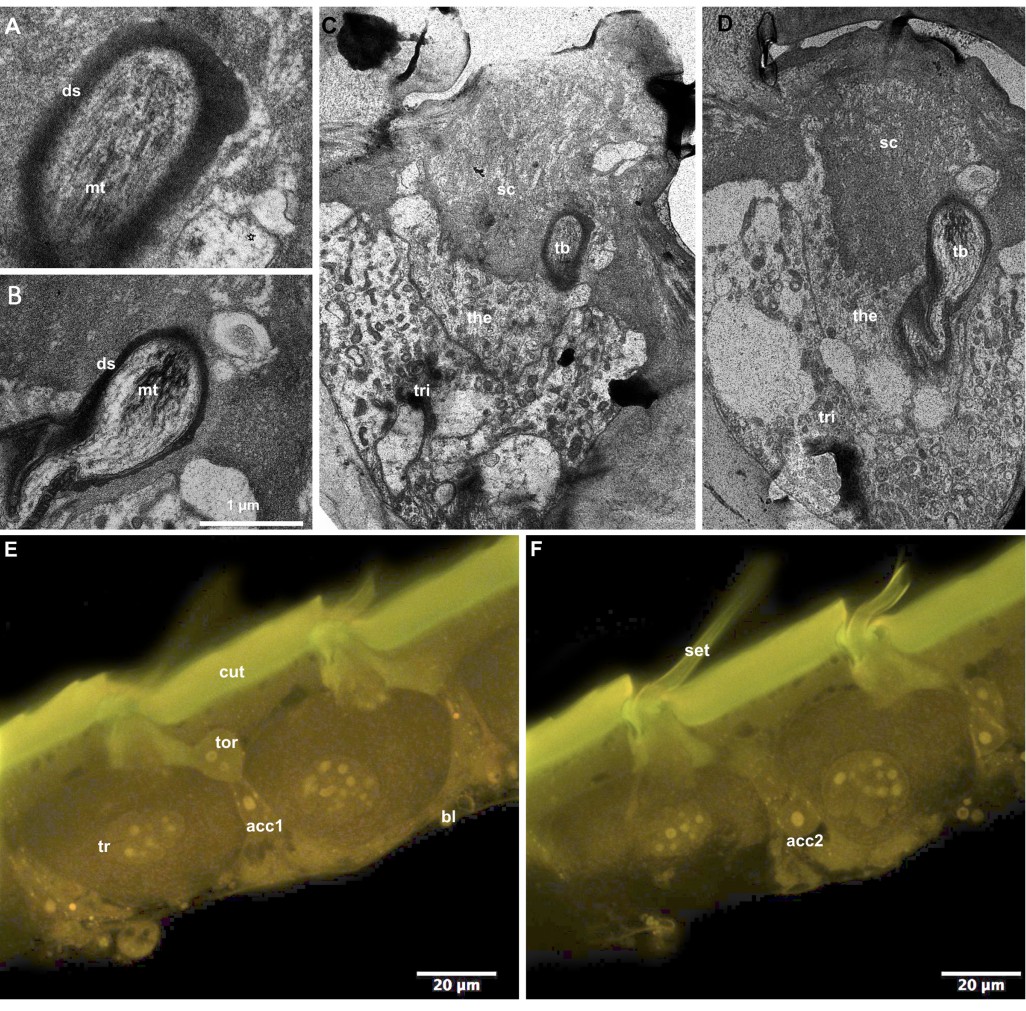

**Figure 5 Cellular pattern of branched setae on T2 in *B. impatiens* ((A–D) TEM, (E and F) CLSM).**
(A and B) Tubular bodies, (C–E) setal bases with adjacent epidermal cells (ds, dendritic sheet; tb, tubular body; mt, microtubules; tri, trichogen cell; the, thecogen cell; cut, cuticle; bl, basal lamina; tor, tormogen cell; acc1, 2, accessory cells; sc, setal cuticle).

## Setal innervation

CLSM observation of the T2 epidermis of callow workers of *B. impatiens* revealed four different cell types associated with setal bases (Figs. 5E and 5F). The distal portion of the largest trichogen cells (tr) are surrounded by the tormogen cell (tor). Two smaller cells are located at the level of the nucleus of the trichogen cells (acc1, acc2). TEM examination of the same type of branched setae on T2 revealed the presence of a single microtubule-rich structure (mt: Figs. 5A and 5B, tb: Figs. 5C and 5D) that is surrounded by electrondense sheets (ds: Figs. 5A and 5B). The structure is embedded in the base of the setal cuticle (sc: Figs. 5C and 5D) and is about 1–2 µm wide. Two cells surround the microtubule containing structure, thecogen cells (the) and trichogen cells (tri).

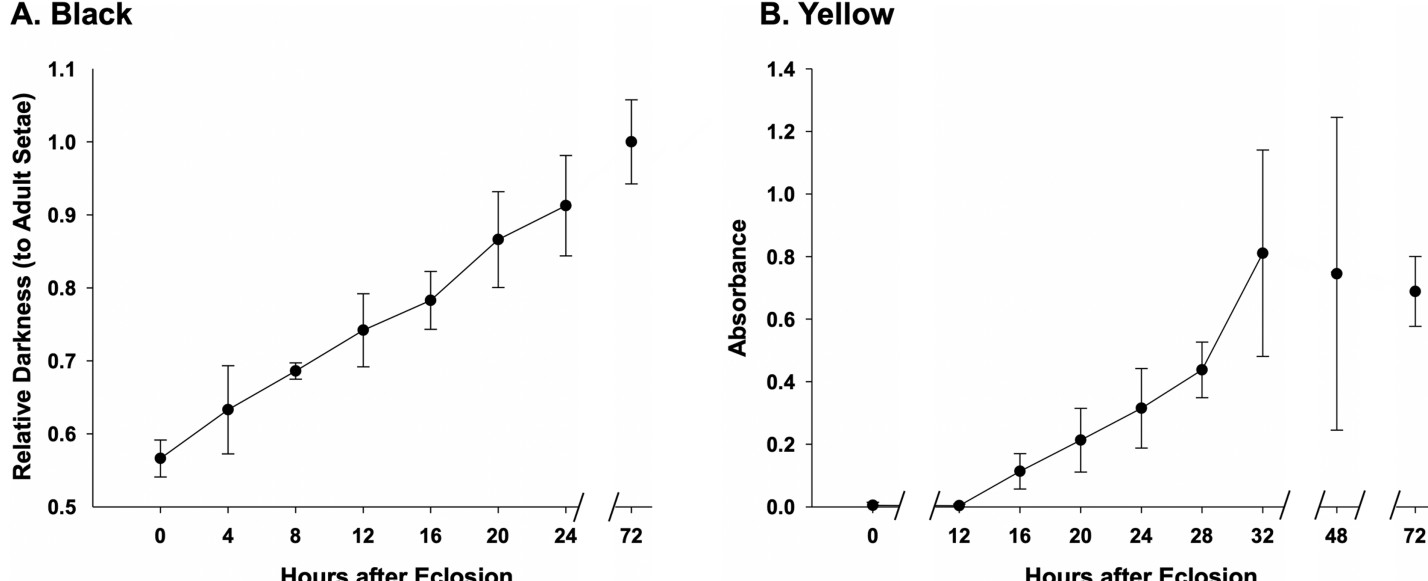

**Figure 6 Temporal pigmentation pattern of cuticular setae in bumble bee *B. impatiens* from 0–72 h post adult emergence "eclosion".** (A) Pigmentation of the black setae is measured as relative darkness of the developing setae to the level of the fully melanized adult (72 h post adult emergence). (B) Pigmentation process of the yellow setae, represented by spectrophotometer absorbance of pigment solution extracted from the developing setae. Note that in the black setae, setal color already reaches 50% of adult intensity at the time of eclosion, and achieves ~90% full adult melanization by 24 h, whereas yellow setae start pigmentation around 12 h post adult emergence and complete pigmentation around 32 h post emergence.

## Setal coloration

Assessment of melanization in black setae (Fig. 6) reveal that setae already obtain 50% of adult color at the time of eclosion and are 90% melanized by 24 h after eclosion (Fig. 6A, Datafile S3). Melanization begins at the root of the setae and gradually reaches the apex (Fig. 2E). Our results suggest that yellow setae start pterin-based pigmentation 12 h post adult emergence from the cocoon, and complete pigmentation is reached by 32 h (Fig. 6B), data which closely match the observed macroscopic shifts in color of these bees. While yellow setae likely contain some melanin as well as the undescribed yellow pterin to impart color, our extractions, which should remove mostly pterin and only some melanin, resulted in the near complete loss of yellow color, leaving setae a light tan color. Melanization is thus ~75% complete by the time pterin pigmentation starts. Black-fated setae appear to be hollow from the point of adult emergence (0 h) to full development, whereas the lumen in yellow setae has a more irregular mottled appearance (Fig. 2D).

## DISCUSSION

In this study we provide initial documentation of the diversity of setal types in bumble bees that serves as a foundation for understanding functioning of these setae and the utility of these traits for clade diagnosis. We reveal that there are two setal types that predominate across bumble bees–the long-branched plumose setae and the short-branched spinulate setae. These setal types can vary by species but are fairly conserved within species across both individuals and caste, and show considerable phylogenetic signal at a more local clade

level. These qualities together make this trait potentially useful for species diagnosis, aiding species delimitation research, and in phylogenetic assessments.

Bumble bees exhibit extensive mimicry, with species from disparate clades converging on coloration. Given that setal color is a primary trait used for species diagnosis and that there is a general paucity of other morphological trait diversity in bumble bees (*Michener, 2000*), it can be difficult for non-experts to discern species. While looking at setal phenotypes requires microscopic examination and a somewhat trained eye, it is yet another trait that can be used to help discern mimics. For example, mimics in the Eastern U.S. forests—*B. sandersoni* Franklin, *B. perplexus* Cresson and *B. vagans* Smith—are very phenotypically similar and thus hard to distinguish even under the microscope (*Milam et al., 2020*). While setal phenotypes are not different enough in *B. sandersoni* and *B. vagans* to distinguish them in our examination of several individuals, *B. perplexus* has considerably more plumose setae. Furthermore, there may be utility for species delimitation. *B. terrestris* Linnaeus from Italy had considerably less plumose setae than *B. terrestris* from Sweden, which aligns with the variation this species can exhibit by ecoregion (*Rasmont et al., 2008*). Similarly, while members of the *B. trifasciatus* Smith species complex (*Hines & Williams, 2012*) generally had a lot of plumose setae, the extent of this from 30–100% on T2 and T3 varied considerably by populations, and *B. haemorrhoidalis* Smith, which has also been debated in species status (*Hines & Williams, 2012*), has variation in amount of plumose setae in posterior segments by geographic region. While we examined only a few individuals to make these assessments, further research on this trait across more individuals could be fruitful for making species decisions, while also helping to inform the function of these setae.

Comparing these setal types across clades aids the ability to assess the function of these different setal types. One such function could be for pollen collection. The face and the mesosomal dorsum are the most plumose parts of the body, showing a very high density of plumosity. These are also the primary parts making contact with the floral surface and the setae that thus hold the most pollen. The lateral regions of the mesosoma and and the lateral margins of the dorsal mesosomal regions—the parts with highest plumosity—are the parts most easily contacted by the legs during grooming. Thus one could argue for the pollen collecting function. However, the socially parasitic *Psithyrus* do not collect pollen and have lost the ability to do so, as observed by the loss of their corbiculae or pollen collecting behaviors, and they instead feed on pollen gathered by their host colonies. This lineage shows no apparent loss of density of plumose setae in the mesosomal and head region, although they do show less metasomal plumosity in general than other long tongued bees along with their general lower density of metasomal setae (*Lhomme & Hines, 2019*). The variation in location of plumose setae across species could relate to the types of flowers that they visit as the longest tongued bees tend to have more metasomal plumosity. Although there could be a phylogenetic component to this, longer tongued bees may visit more tubular flowers, such as legumes, and thus we might expect that these bees make more metasomal contact with floral surfaces than those likely to visit shorter more open flowers.

Another function of these setae may be to aid heat retention in the metasoma in cold areas. The mesosoma is the furnace of the bee, generating the heat needed for the body and for flight and thus the ability to retain heat in the heat generator using plumose insulation could be a function of these setae. The metasoma in contrast serves to regulate heat dissipation thus variation in plumosity could relate to the need to dissipate or retain heat. In further support of a thermoregulatory role, *Alpinobombus* is the most plumose of the short-tongued clade and these are the species that occur in the coldest regions. Similarly, some of the warmer adapted long-tongued bees, such as the New World *Thoracobombus*, have the least plumose setae on their metasomas, and within clades like *Bombus* s.s., the colder adapted species (*e.g.*, *patagiatus*, *moderatus*) and populations (*e.g.*, *terrestris*) have more plumosity. Nevertheless, there are exceptions to this, as the subtropical SE Asian comimics *trifasciatus*, *haemorrhoidalis*, and *breviceps* Smith have high plumosity and the sister lineage to the remaining bumble bees—*Mendacibombus*—are high altitude species, but yet have the least plumosity in the metasoma and overall. This could be driven by phylogenetic constraints, however, as data support the increase in plumosity did not occur in the basal node of bumble bees but increased with the primary radiation. There could potentially be a role in water retention as well, such as the ability to prevent rainwater from soaking into the pile and preventing flight. *Bombus trinominatus* has considerable plumosity compared to sister lineages and tends to fly in misty habitats (*Labougle, 1990*), and the SE Asian comimics above occur in wet habitats. A better understanding and thorough analysis of habitat use by bumble bee species is needed to assess these trends more fully.

Another potential function of these setae is mechanosensory. Bumble bees can perceive electrostatic signals of flowers (*Clarke et al., 2013*), and setae of the pile on the head and mesosoma may play a role in this function (*Sutton et al., 2016*). Plumose setae may have differing electrostatic properties that aid floral learning or handling or could be mechanosensory in themselves. The presence of high plumosity in the face around the antennae would support such a function (although this could also promote heating of the head/brain). Also, there often is a row of shorter plumose setae and other smaller setal types only on the tergite margin. These have the appearance of being mechanosensory in function given that they are too small and not dense enough to serve other purposes, and the distal location on the tergite suggests this region is likely to be important for perceiving external signals or interacting with other body parts. Our data examining the marginal branched setae of *B. impatiens* revealed a cellular pattern consistent with these being innervated. Supporting this is the presence of smaller accessory cells resembling the thecogen cells and/or the cell bodies of sensory neurons (acc1, acc2: Figs. 5E and 5F) and a microtubule rich structure with electrondense sheet (ds) that strongly resembles the tubular body. Further research is needed on whether the different seta types may vary in innervation, which would be needed for mechanosensation.

In addition to the main two seta types, there are also other seta types confined to specific locations on the bee. Notably, there are unbranched simple setae on the bees but usually in locations unlikely to be of use for pollen collection and/or in which there is more need for heat dissipation. Simple setae and other shorter setal types (fine pectinate, dendritic) are

abundant on the metasomal sternites (Fig. 2C), which are used to dissipate heat from females to brood during incubation. One of the greatest differences of males from females is in having denser, longer, and more branched setae in the sternites, which could relate to the lack of need for heat dissipation to brood, and need for improvement on insulation. Some of these setae could also play roles in mechanosensation during mating. Simple setae also occur in the anterior tergites in the region where tergites overlap and thus do not make contact with pollen. Simple and smaller setae could also prevent accumulation of wax in setae, which is produced in the anterior portions of sternites and tergites near where these occur (*Landim, 1963*). The location of bipentunculate setae confined to just posterior to the wing may serve a purpose in interacting with wings. Examination of setae of the legs shows a wide diversity of setal types by location that likely relates to various grooming functions of setae. Further analysis of these other setal types, especially as it relates to potential mechanosensory roles and grooming behaviors, is needed across bumble bee species.

Bumble bee setae also impart adaptive colors. We did not see any relationship between seta type and color for the two major seta types, although some of the smaller setae did not match the color of the larger, showier setae in their respective regions, suggesting that the color may be less important to their function. Unlike what has been observed in scales in some butterfly species (*e.g.*, *Janssen, Monteiro & Brakefield (2001)*), development of setal morphologies complete around pupal stage P7, considerably earlier than the start of setal pigmentation around pupal stage P13 (*Tian & Hines, 2018*), thus morphological development is decoupled from color. Our developmental data show that the yellow and black setal colors are temporally separated in their timing in early adults, with most of the melanin deposited by the time yellow pterin-like pigmentation begins. As differentiation between melanized red or black and predetermined yellow setae is apparent by later pupal stages, the first genetic change for yellow setae is loss of melanization and subsequently they acquire their yellow pigments. The setae show different appearance in the lumen suggesting the different ways in which these colors may be regulated molecularly, with the appearance of more fluid in yellow setae concordant with the later deposition of the yellow pterin-like pigment in these setae. These may suggest that changes in cellular control of lumen contents could be involved in this regulation. These data also show that while these bees appear to be fully melanized by 12–24 h post adult emergence, there is some additional melanization that occurs thereafter.

Our setal development data also provide a timeline for examining the development and evolution of setal types useful for future research, and which can serve as a model for better understanding cellular branching processes in insects (*Deans et al., 2015*). The rachis develops first, followed by formation of blunt ended branches, followed by distal extension of the rachis for final separation of distal branches, and subsequently the branches become pointed around the time that cuticle is deposited. Different seta types may thus be influenced by initial location of branch formation along the seta (pectinate *vs* spinulate *vs* simple), degree of rachis extension (*e.g.*, dendritic), rates of branch growth (plumose *vs* spinulate), and the degree to which points develop in the final stages (blunter short spinulate).

## CONCLUSIONS

Our study documents the diversity of setal morphologies across the characteristically densely pilous bumble bee body. It reveals considerable variation in more plumose setae across this lineage that suggests roles for both pollen collection and thermoregulation. The study favors mechanosensory roles of at least some of the smaller setae and suggests a need to better understand the function of these setae. Finally, it raises setal morphologies as potential diagnostic and phylogenetic traits for future systematic research on these bees. Future research in this area should focus more on mechanosensory functions and apply comparative analysis of habitat regimes across the clade to better understand setal function.

## ACKNOWLEDGEMENTS

We would like to thank Sydney Cameron for assisting and supporting SEM images of setae and providing specimens for analysis, Shiyu Zha for taking SEM images of leg setae, and Missy Hazen from the Huck Microscopy Facility for assistance with imaging.

### Funding

This work was supported by the National Science Foundation Division of Environmental Biology grants 1453473 and 2126417. S. Kilpatrick was supported by a US Department of Agriculture NIFA NNF Graduate training fellowship (#2017-38420-26766). The funders had no role in study design, data collection and analysis, decision to publish, or preparation of the manuscript.

### Grant Disclosures

The following grant information was disclosed by the authors:
National Science Foundation Division of Environmental Biology: 1453473 and 2126417.
US Department of Agriculture NIFA NNF Graduate Training Fellowship: 2017-38420-26766.

### Competing Interests

The authors declare that they have no competing interests.

### Author Contributions

- Heather M. Hines conceived and designed the experiments, performed the experiments, analyzed the data, prepared figures and/or tables, authored or reviewed drafts of the article, and approved the final draft.
- Shelby Kerrin Kilpatrick conceived and designed the experiments, performed the experiments, analyzed the data, prepared figures and/or tables, authored or reviewed drafts of the article, and approved the final draft.

- István Mikó conceived and designed the experiments, performed the experiments, analyzed the data, prepared figures and/or tables, authored or reviewed drafts of the article, and approved the final draft.
- Daniel Snellings conceived and designed the experiments, performed the experiments, analyzed the data, prepared figures and/or tables, and approved the final draft.
- Margarita M. López-Uribe conceived and designed the experiments, performed the experiments, analyzed the data, prepared figures and/or tables, authored or reviewed drafts of the article, and approved the final draft.
- Li Tian conceived and designed the experiments, performed the experiments, analyzed the data, prepared figures and/or tables, authored or reviewed drafts of the article, and approved the final draft.

## Field Study Permissions

The following information was supplied relating to field study approvals (*i.e.*, approving body and any reference numbers):

All specimens used for the study were obtained for other research studies and thus were pulled from prior collections and published research (*e.g.*, most bumble bees used were vouchers from Cameron et al., 2007) OR were obtained locally at sites not requiring permits (University property, the authors' personal property) OR were obtained from purchased bumble bee colonies (Koppert, citations provided in text). No permits were thus required for any material acquired specifically for this study.

## Data Availability

The raw data is available in the Supplemental Files.

## Supplemental Information

Supplemental information for this article can be found online at http://dx.doi.org/10.7717/peerj.14555#supplemental-information.

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
