# Peer review of "The diversity, evolution, and development of setal morphologies in bumble bees (Hymenoptera: Apidae: Bombus spp.)"

_PeerJ, doi:10.7717/peerj.14555_

## Round 0.1 · original submission · Minor Revisions

I believe all reviewer comments should be addressed. In particular, please pay careful attention to the comments by Reviewer 1 on terminology/nomenclature. This reviewer likes the paper but strongly wishes for good terminology, as the paper will set the standards in this area going forward.

Reviewer 1 ·

Basic reporting

See attachment.

Experimental design

See attachment.

Validity of the findings

See attachment.

Annotated reviews are not available for download in order to protect the identity of reviewers who chose to remain anonymous.

Reviewer 2 ·

Basic reporting

This review pertains to The diversity, evolution, and development of setal 1 morphologies in bumble bees (Hymenoptera: Apidae: Bombus spp.) (#77797) submitted to PeerJ. The manuscript is a detailed descriptive paper detailing for the first time the different types (and subtypes) of bumble bee pile/setae/hair morphology and its distribution across different parts of the bumble bee body plan. The paper has several components:
1) define all setal morphologies using a single species, comparing individuals within and between social castes using microscopy of hairs shaved from bees.
2) using the well-resolved phylogeny of Bombus, compare the body distribution of the most common types of setal morphology among species and clades using comparative phylogenetic methods.
3) examine setal development: branching patterns using compound microscopy, innervation using CLSM and TEM, and pigmentation timing using spectrophotometry.

Overall I was very interested in this paper and results given the number of hypotheses relating to possible roles and evolutionary pressures acting on setal morphology. The methods and results are described in very nice detail, and the incorporation of all three elements, especially the phylogenetic comparison, made the paper a very comprehensive read(even if preliminary for some areas, but thats not a bad thing here). I only have minor comments to improve clarity or fix typos, largely in methods and results and figures. I will present these points in order of manuscript section.

The writing was clear and detailed, the relevant literature was cited. The relevant data appears to have been uploaded.

Experimental design

I have no major comments related to methods or experimental design. All comments are minor.
Methods:
L181: The "Setal Morphology" sections is largely fine, and focuses on B. impatiens. The authors present detailed SEM imagery of major setal types come from B. bimaculatus however. The species are closely related and I doubt there are many differences between the two species, it just was surprising that the images weren't from impatiens here. Some rationale for using the other species when impatiens was the focus of this section should be added in, even if it's just "these are closely related and we had existing SEM images for this species and they look the same".


L238: Add abbreviation for CLSM here at first usage.
L250: TEM will be defined in Intro (L110) so abbreviation can be used here.
L251-253. I feel like this sentence (starting with "For CLSM, specimens where..." is missing some words. Please rewrite or clarify.
L280: Are these time points the same for black and yellow? Were different time points used for black? Please clarify the sampling points for black.
L292: for the absorbance correction equation, could the authors clarify what a "length unit" is?

Validity of the findings

The findings are largely descriptive but very useful to researchers of bumble bees and their evolution. The fact that there is phylogenetic signal in characters could be useful for taxonomists as well.
No major concerns relating to the validity of findings and conclusions derived from these findings. Statistcal analyses are limited given the descriptive nature of most aspects of the paper, but appear to be valid where needed (e.g., in the comparative phylogenetic analysis, temporal analysis of pigmentation, etc).

The following comments related to the Results and figures.
Figure 1 presents major setal types and subtypes as well as body plan locations. I had a bit of a hard time navigating this figure. I recommend strongly increasing the black line weight between major panels (between the Major Types), or perhaps better, separating them with white space similar to Figure 4. I just had to look for quite a long time before figuring out which images went with each major type. Same could probably be said for Figure 2 and 5.

In discussion L632 the authors state that they did not see any relationship between hair type and color, but I could not find any formal analysis of this in Results. It's possible I am overlooking a statement relating to color/morphology somewhere, but I think it would perhaps be warranted to have a few sentences describing how color is unrelated to structure. This could probably be addressed in all three sections, but most obviously in the third part of the paper. Some discussion or examples of pigment variation within clades and how it relates to the setal morphology for the phylogenetic section might also be interesting. I don't think at this stage that any statistical comparisons would be needed however, and would be more the focus of a followup study.

---

## Round 0.2 · accepted · Accept

The authors have made most of the changes suggested by reviewers and have provided justifications when no changes were made. The paper is ready for publication.